# Research on Features of Pipeline Crack Signal Based on Weak Magnetic Method

**DOI:** 10.3390/s20030810

**Published:** 2020-02-02

**Authors:** Bin Liu, Ziqi Liu, Ning Luo, Luyao He, Jian Ren, He Zhang

**Affiliations:** College of Information Science and Engineering, Shenyang University of Technology, Shenyang 110870, China; liubinsgd@sut.edu.cn (B.L.); ilccmm@163.com (N.L.); tgx199515@163.com (L.H.); yy085638@163.com (J.R.); zhanghe6501273@163.com (H.Z.)

**Keywords:** weak magnetic, magnetic charge, signal characteristic, crack

## Abstract

Quantitative online detection of microcracks in long-distance oil and gas pipelines is an international problem, and the effective detection method is still lacking. In this paper, a mathematical model of non-uniform distribution of crack magnetic charges is established based on the stress distribution laws of pipeline cracks under internal pressure. The weak magnetic signal characteristics of pipeline cracks with different sizes are analyzed. The internal pressure increasing factor of weak magnetic signals are extracted to analyze the corresponding relationship between crack size and weak magnetic signals. The experimental study of the X70 pipeline is carried out. The results show that the axial component of the weak magnetic signal at the crack has a maximum value near the tip, and a minimum value appears in the middle of the crack. The internal pressure increasing factor is introduced to quantify the weak magnetic signal, the crack is in a safe state (not expanding) when the internal pressure increasing factor is positive, the weak magnetic signal has a linear relationship with the crack size. However, the crack is in a dangerous state when the internal pressure increasing factor is negative, and the pipeline crack will expand as the internal pressure increases.

## 1. Introduction

Due to the unreasonable arrangement of pipeline structure or defects in materials, stress concentration will be formed in the long-distance oil and gas pipelines, and stress cracks will often occur in the relevant areas, which will affect the stability of the pipeline operation [1,2,3,4,5]. Finally, it will cause serious damage. Conventional NDT methods, such as magnetic flux leakage, magnetic particle, eddy current, permeating, ultrasonic, radiographic, etc., have played an important role in pipeline defect detection and accident prevention [6,7,8,9,10,11,12]. However, pipeline internal inspection technology has non-contact, high-speed detection (the operating speed of the internal detector is 1–5 m/s), dynamic continuous and other technical requirements. At present, magnetic flux leakage detection technology is the most widely applied and most mature pipeline internal inspection technology because it meets the above requirements. Magnetic flux leakage detection technology is adopted by more than 90% of the pipeline internal detection equipment in the world. But there are limitations to the safety assessment of microcracks in pipeline magnetic leakage detection technology. The main reason is that the size of the microcracks is small, resulting in a weak magnetic flux leakage signal generated by the microcracks. And the magnetic flux leakage signal is easily covered by the exciting field of the excitation device, which makes the microcracks difficult to identify [13,14,15,16,17]. 

The principle of weak magnetic microcracks detection is that the stress concentration degree of the in-service pipeline crack is detected to determine the size characteristics of microcracks in the geomagnetic field. The weak magnetic method has the characteristics of non-contact, fast acquisition of signals, support for dynamic detection, and no excitation structure. It has practical significance to improve flexibility and reduce the cost of the inspection in oil and gas pipelines [18,19]. However, weak magnetic signal characteristics can only be used to judge the position of the crack. There is always a lack of methods for the quantitative determination of cracks [20,21].

In practical engineering applications, the stress concentration near the crack differs greatly under the action of internal pressure. Generally, the stress at the crack tip and the crack bottom is larger, while the stress at the crack side surface is smaller. The existing theoretical model calculates the magnetic signal on the premise that the stress concentration near the crack is approximately uniform. Although this method simplifies the calculation process and reduces the amount of calculation, the error is large and it is difficult to accurately describe the weak magnetic signal characteristics of the crack. In this paper, a numerical model of non-uniform crack magnetic charges is established based on the stress distribution laws of pipeline cracks under internal pressure. According to the variation of the weak magnetic signal of pipeline crack with internal pressure, the slope of the radial peak with the change of internal pressure is exacted to be the crack internal pressure increasing factor, which is introduced as the characteristic value to quantify the crack. The relationship between the internal pressure increasing factor and crack size is analyzed. Changes of internal pressure increasing factor before and after crack propagation are analyzed. The systematic experiment is carried out, which lays a foundation for the quantitative detection of microcracks.

## 2. Numerical Model of Non-Uniform Crack Magnetic Charge

In the geomagnetic environment, the magnetic field *H_σ_* generated by the stress can be equal to the applied external magnetic field when the material is subjected to external stress, which can be expressed as [22]:(1)$$H_{\sigma }=\frac{3\sigma }{2\mu_{0}}(\frac{d\lambda }{dM})(\cos^{2}\theta -v^{2}\sin\theta )$$
(2)$$H_{e}=H+\alpha M+H_{\sigma }$$
where, *α* is a coupling parameter, *H* is the external magnetic field, *σ* is applied stress, *λ* is the magnetostrictive coefficient, *M* is the magnetization. *µ_0_* is the vacuum permeability. *θ* is the angle between the directions of the stress and the *H_e_*, *H_e_* is the magnetic field of the material, and *v* is the Poisson’s ratio.

The non-hysteresis magnetization *M_an_* is expressed as:(3)$$M_{an}=M_{s}(\coth(\frac{H_{e}}{a})-\frac{a}{H_{e}})$$
where, *M_s_* is the saturation magnetization, and a is a constant [23].
(4)$$\frac{dM_{irr}}{dW}=\frac{1}{\xi }(M_{an}-M_{irr})$$
where, *ξ* is a constant, which is related to the energy per unit volume, *M_irr_* is the irreversible component of the magnetization. The derivative of the magnetization to the stress energy *W* is expressed as:(5)$$\begin{array}{cc} \frac{dM}{dW} & =\frac{\left(1-c\right)}{\xi } \end{array}(M_{an}-M_{irr})+c\frac{dM_{an}}{dW}$$
where, *c* is the reversible coefficient, that is, the parameter of the reversible motion of the magnetic domain. While, the derivative of stress energy is expressed as:(6)$$dW=(\frac{\sigma }{E})d\sigma$$
where, *E* is the elastic modulus, substituting the Equation (6) into the Equation (5) to obtain the relationship between magnetization and stress as following [24]:(7)$$\frac{dM}{d\sigma }=\frac{1}{\varepsilon^{2}}\sigma \left(1-c\right)\left(M_{an}-M_{irr}\right)+c\frac{dM_{an}}{d\sigma }$$
where, *ε* is a constant. If there is an axial outer surface crack on the pipeline, and the crack size is *2C* long, the width is *b*, and the depth is *d*, the stress *σ* at the crack tip is obtained [25]:(8)$$\sigma =\frac{F\sqrt{\pi d/Q}pR}{t}$$
where, *Q* is an expansion coefficient, which can be express as:(9)$$Q=\sqrt{1+1.61(d^{2}/Rt)}$$

*F* is the stress intensity factor at the crack tip, which is only related to the crack size and the internal pressure. *p* is the internal pressure, *R* is the pipeline radius, *t* is pipe wall thickness, *d* is the crack depth.

It can be seen from Equation (8) that the stress at the crack tip of the pipeline is only related to the internal pressure and the crack size. Since the stress concentration distribution at the crack is not uniform under the load, the magnetic charge at the crack is accumulated at the crack tip, and the distribution is less on the side, thereby forming an internal magnetic source.

Because the internal stress distribution of the crack is not uniform, the magnetization of the interior will be different. The relationship between the magnetic charge density ρ and the magnetization M is expressed as [26]:(10)$$\rho =MtD\mu_{0}$$
where, *D* is the diameter of the pipeline, and *t* is the wall thickness, and the magnetic charge density throughout the crack is as shown in Figure 1.

Assuming that the center of the bottom of the crack is the origin of the coordinate, the magnetic field strength generated by the single magnetic charge at the spatial point is:(11)$$dH=\frac{\rho rdydz}{2\pi \mu_{0}{\left|r\right|}^{3}}$$
where, *r* is the distance from the spatial point to the origin.

Due to the correspondence between the magnetic charge density and the magnetization, the magnetic charge distribution formula can be obtained by the stress distribution, and *ρ*_1,_
*ρ*_2_ are magnetic charge density at the crack tip and crack side, respectively.
(12)$$\rho_{1}=\rho_{max}\frac{m}{d}z+(1-\frac{m}{d})\rho_{max}$$
(13)$$\rho_{2}=(1-\frac{m}{d})\rho_{\max}-(\frac{n+\frac{m}{d}-1}{d}z)\rho_{\max}$$
where, *m*, *n* are constants. The weak magnetic signal generated by the crack tip at point (*x*_0_, *y*_0_, *z*_0_) can be obtained:(14)$$H_{r}=\int_{0}^{b}dy\int_{0}^{d}\frac{\rho_{\max}(\frac{m}{d})z+(1-\frac{m}{d})\rho_{\max}}{2\pi \mu_{0}\left[{\left(x-x_{0}\right)}^{2}+{\left(y-y_{0}\right)}^{2}+{\left(z-z_{0}\right)}^{2}\right]}(z_{0}-z)dz$$
(15)$$H_{a}=\int_{-C}^{C}dx\int_{0}^{d}\frac{\rho_{\max}(\frac{m}{a})z+(1-\frac{m}{a})\rho_{\max}}{2\pi \mu_{0}\left[{\left(x-x_{0}\right)}^{2}+{\left(y-y_{0}\right)}^{2}+{\left(z-z_{0}\right)}^{2}\right]}(x_{0}-x)dz$$
where, *H_r_* is the radial component of the weak magnetic signal, *H_a_* is the axial component of the weak magnetic signal. The weak magnetic signal generated by the both sides of the crack at point (*x*_0_, *y*_0_, *z*_0_) can be obtained:(16)$$H_{r}=\int_{-C}^{C}dx\int_{0}^{d}\frac{(1-\frac{m}{d})\rho_{\max}-(\frac{n+\frac{m}{d}-1}{d}z)\rho_{\max}}{2\pi \mu_{0}\left[{\left(x-x_{0}\right)}^{2}+{\left(y-y_{0}\right)}^{2}+{\left(z-z_{0}\right)}^{2}\right]}(z_{0}-z)dz$$
(17)$$H_{a}=\int_{-C}^{C}dx\int_{0}^{d}\frac{(1-\frac{m}{d})\rho_{\max}-(\frac{n+\frac{m}{d}-1}{d}z)\rho_{\max}}{2\pi \mu_{0}\left[{\left(x-x_{0}\right)}^{2}+{\left(y-y_{0}\right)}^{2}+{\left(z-z_{0}\right)}^{2}\right]}(x_{0}-x)dz$$

The available weak magnetic signals for the crack model in the axial and radial directions are expressed as:(18)$$\begin{array}{ll} H_{r} & =\int_{0}^{b}dy\int_{0}^{d}\frac{\rho_{1}}{2\pi \mu_{0}r_{1}{}^{2}}(z_{0}-z)dz+\int_{0}^{b}dy\int_{0}^{d}\frac{\rho_{1}}{2\pi \mu_{0}r_{2}{}^{2}}(z_{0}-z)dz\\ & \;+\int_{-C}^{C}dx\int_{0}^{d}\frac{\rho_{2}}{2\pi \mu_{0}r_{3}{}^{2}}(z_{0}-z)dz+\int_{-C}^{C}dx\int_{0}^{d}\frac{\rho_{2}}{2\pi \mu_{0}r_{4}{}^{2}}(z_{0}-z)dz \end{array}$$
(19)$$\begin{array}{ll} H_{a} & =\int_{0}^{b}dy\int_{0}^{d}\frac{\rho_{1}}{2\pi \mu_{0}r_{1}{}^{2}}(x_{0}-x)dz+\int_{0}^{b}dx\int_{0}^{d}\frac{\rho_{1}}{2\pi \mu_{0}r_{2}{}^{2}}(x_{0}-x)dz\\ & \;+\int_{-C}^{C}dx\int_{0}^{d}\frac{\rho_{2}}{2\pi \mu_{0}r_{3}{}^{2}}(x_{0}-x)dz+\int_{-C}^{C}dx\int_{0}^{d}\frac{\rho_{2}}{2\pi \mu_{0}r_{4}{}^{2}}(x_{0}-x)dz \end{array}$$
where, *r*_1_, *r*_2_, *r*_3_, *r*_4_ are the distances between the crack tip and the side of the crack to the detection point (*x*_0_, *y*_0_, *z*_0_), respectively.

Taking the X70 of large-diameter long-distance oil and gas pipeline application as an example, the outer diameter of the pipe is 1012 mm, the wall thickness is 14.5 mm, and the crack size is 30 mm long, 1 mm wide, and 3 mm deep. When the pressure of the pipe is 3 MPa, the crack signal characteristic of the non-uniform magnetic charge is shown in Figure 2.

## 3. Calculation of the Simulation

To obtain the quantitative relationship between crack size and weak magnetic signal, a numerical simulation model of magneto-mechanical coupling is established in this paper. The length of the pipeline model is 5000 mm, the wall thickness is 14.5 mm, the pipe diameter is 1016 mm, and the crack is located at the center of the pipeline surface, which is 30 mm long, 3 mm deep, and 1 mm wide. Axial displacement constraints are applied to both sides of the pipeline model to simulate pipeline operation. The internal pressure of 3 MPa is applied to the inner wall of the pipeline, and the linear elasticity is calculated. The stress diagram at the crack is shown in Figure 3.

It can be seen from Figure 3 that the maximum stress of the crack occurs at the crack tip. The stress at the crack tip section decreases as the depth increases, and the stress on both sides of the crack increases as the depth increases.

In the model space shown in Figure 3, the uniform magnetic field strength is set to 50 nT to simulate the geomagnetic field environment. The characteristics of the weak magnetic signal at the crack are simulated as shown in Figure 4.

It can be seen from Figure 4 that the axial component of the weak magnetic at the crack has a maximum value near the crack tip, and a minimum value appears in the middle of the crack. The radial component of the weak magnetic at the crack has a peak and a valley at the crack tip. In order to study the characteristics of pipeline crack weak magnetic signals under different internal pressures, the internal pressures of 1 MPa~3 MPa are applied to the pipeline, and the weak magnetic signals and stresses at the crack are evaluated at an interval of 0.5 MPa, the stress changes at the crack tip are shown in Figure 5.

As seen in Figure 5, the stress at the pipeline crack tip increases linearly with the increase of the internal pressure. The magnetization of the crack increases as the stress at the crack increases, and the characteristic change of the weak magnetic signal at the crack is shown in Figure 6.

It can be seen from Figure 6 that the characteristics of the weak magnetic signal at the crack vary with the internal pressure, and the characteristics of the weak magnetic signal at the pipeline crack change linearly with the internal pressure. That is, the axial maximum and the radial peak of the weak magnetic signal at the crack increase linearly with the increases of internal pressure.

Since the tip signal at the crack is most sensitive to crack signal changes, the slope of the radial peak with the change of internal pressure is exacted to be the internal pressure increasing factor *k*. 

In order to study the influence of crack depth on the weak magnetic signal of pipeline cracks, the characteristics of crack signals at different depths are obtained by simulation. The crack size is 30 mm long and 1 mm wide, the depth range is 1~3 mm, and the interval is 0.5 mm. The stress change at the crack tip with the depth is shown in Figure 7.

As seen in Figure 7 that the stress at the pipeline crack tip also increases linearly as the pipeline crack depth increases. The magnetization at the crack increases as the stress at the crack increases. The characteristics of the weak magnetic signal at the crack change with the crack depth are shown in Figure 8a,b.

It can be seen from Figure 8a,b that the weak magnetic signal characteristics of the pipeline crack change with the crack depth. And the weak magnetic axial maximum and the radial peak at the pipeline crack are extracted. The crack depth is linearly related to the weak magnetic signal at the pipeline crack, that is, the weak magnetic axial maximum and the radial peak at the crack increase linearly with the crack depth.

In order to compare and analyze the effect of crack depth on the characteristic value of the crack signal, the peak-to-peak values of the radial weak magnetic signal with different depths are extracted, which can be shown in Figure 9.

As seen in Figure 9 that the Radial peak-to-peak values of the weak magnetic signal with different depth cracks change linearly with the internal pressure. The internal pressure increasing factor *k* of different crack sizes are calculated. The internal pressure increasing factor *k* of the weak magnetic signal at the crack is linear with the crack depth. The relationship between the internal pressure increasing factor *k* and the crack depth *d* is obtained as follows:(20)k=f·d+g
where, *f* and *g* are constants.

In order to study the influence of crack length on the weak magnetic signal of pipeline crack, the simulation analysis of cracks with different lengths is carried out. The crack size is 3 mm in depth and 1mm in width, the crack length range is 20~40 mm, and the interval is 5 mm. The stress change at the crack tip under the different lengths is shown in Figure 10.

It can be seen from Figure 10 that the stress at the crack tip increases linearly as the crack length increases, and the magnetization at the crack also increases as the stress increases. The relationship between the weak magnetic signal characteristics and the crack length at the crack is shown in Figure 11a,b.

It can be seen from Figure 11a,b that the weak magnetic signal characteristics of the pipeline crack change with the crack length. And the weak magnetic axial maximum and the radial peak at the crack increase linearly with the crack length.

In order to compare and analyze the effect of crack length on the characteristic value of the crack signal, the peak-to-peak values of the radial weak magnetic signal with different lengths are extracted, which are shown in Figure 12.

It can be seen from Figure 12 that the weak magnetic radial peak signal at the crack changes linearly with the internal pressure. The internal pressure increasing factor *k* of the weak magnetic signal at different cracks is calculated. That is, the internal pressure increasing factor *k* of the weak magnetic signal at the crack also has a linear change trend with the crack length. And the influence of depth and length on the internal pressure increasing factor *k* of the weak magnetic signal at the crack is compared. The crack depth has a greater influence on the internal pressure increasing factor *k* of the weak magnetic signal at the crack.

When the crack is in a dangerous state, the weak magnetic signal at the crack decreases with the internal pressure increases because of the large plastic deformation at the crack. That is, the characteristic value of the weak magnetic signal at the crack decreases as the internal pressure increases. The internal pressure increasing factor of the weak magnetic signal at the crack is less than 0, which can be shown as:(21)k<0

The internal pressure increasing factor of the weak magnetic signal at the crack is negative.

## 4. Experiment

In order to verify the correctness of theoretical calculations and simulation results, a pipeline pressure test is designed in this paper. The experimental material is X70 pipeline (6 m long) prefabricated with artificial cracks, with a diameter of 1016 mm and a wall thickness of 14.5 mm. Weak magnetic signals of different crack sizes are detected under different pressures during the experiment. Magnetic signal acquisition is performed using a weak magnetic detection device with an accuracy of 0.001 A/mm. In the pipeline experiment, the internal pressure range is 0~3 MPa, and the weak magnetic signals of the pipeline crack are detected at intervals of 0.5 MPa. Crack numbers of different sizes in the experiment are shown in Table 1. The experimental pipeline and weak magnetic detection device are shown in Figure 13.

To study the characteristics of weak magnetic signals at the pipeline cracks under internal pressure, the weak magnetic signal of the No. 3 crack is extracted under the pressure of 3 MPa as shown in Figure 14.

It can be seen from Figure 14 that the axial component has a maximum value near the crack tip, and a minimum value appears in the middle of the crack. The radial component has a peak and a valley at the crack tip. The experimental error has been shown in the figure with an error bar. Comparing the experimental and simulation results, the characteristics are highly consistent. However, there are certain errors in the initial cracks made during the experiment, and impact factors, such as fluctuations in the water pressure of the pipeline and different initial magnetization states of the pipeline, will affect the detection results.

In order to further study the relationship between the crack signal and the internal pressure, the weak magnetic signals at the crack under different internal pressures are extracted, which is shown in Figure 15.

In order to study the influence of crack depth on the weak magnetic signal characteristics of cracks under internal pressure, the weak magnetic signals of cracks with different depths are extracted. The characteristic changes of crack signals of different depths are shown in Figure 16.

It can be seen from Figure 16 that the pipeline crack depth under the internal pressure is deeper, the weak magnetic signal characteristic at the crack is more obvious. To further compare and analyze the influence of crack depth on the characteristics of the weak magnetic signal, the radial peak-to-peak values of weak magnetic signals with different depth cracks are extracted, which is shown in Figure 17.

As seen in Figure 17 that the radial peak-to-peak of weak magnetic signal increases at the crack as the internal pressure increases, and the internal pressure increasing factor *k* of the different cracks are obtained by calculation. The relationship of the internal pressure increasing factor *k* and the crack depth is obtained, which is shown in Figure 18.

It can be seen from Figure 18 that the internal pressure increasing factor *k* at the crack increases as the crack depth increases. And the internal pressure increasing factor *k* at the crack has a linear relationship with the crack depth. The formula can be obtained by fitting:*k* = 0.7*d* + 2.37(22)

The same method is used to calculate the internal pressure increase factors *k* for cracks of different lengths, which is shown in Figure 19.

It can be seen from Figure 19 that the internal pressure increasing factor *k* at the crack increases as the crack length increases. And the internal pressure increasing factor *k* at the crack has a linear relationship with the crack length. The formula can be obtained by fitting:*k* = 0.08*C* + 1.27(23)

Therefore, it can be seen that the weak magnetic signal at the crack has a linear relationship with the internal pressure when the crack does not start to expand. Because the crack size has a linear relationship with the weak magnetic signal at the crack, the crack size can be effectively evaluated by the internal pressure increasing factor *k*. 

To compare and analyze the changes of the weak magnetic signal before and after the crack expands, a pipeline pressure test is performed on the pipeline. During the experiment, the weak magnetic detection device and the strain gauge are attached to the crack tip to monitor the stress and the weak magnetic signal at the crack tip. When the crack begins to expand under the action of internal pressure, the change of the weak magnetic signal at the crack is shown in Figure 20. That is, as the stress increases, the weak magnetic signal at the crack decreases. The internal pressure increasing factor *k* of the weak magnetic signal at the crack is negative. Thus, the internal pressure increasing factor *k* can effectively determine if the crack will expand.

## 5. Conclusions

In this paper, a non-uniform magnetic charge model is established based on the stress distribution laws of pipeline cracks under internal pressure. The corresponding relationship between the crack size and the weak magnetic signal of the crack is quantitatively analyzed. The signal characteristics at the crack are extracted to determine the safety of the crack. The results show that: the axial component of the weak magnetic signal at the pipeline crack has a maximum value near the tip, a small value appears in the middle of the crack, and the radial component has a peak and a valley at the tip. The slope of the radial peak with the change of internal pressure is exacted to be the crack internal pressure increasing factor, which is mainly affected by the crack size. When the internal pressure increasing factor of the crack is positive, the crack is in a safe state, and the internal pressure increasing factor *k* of the crack has a linear relationship with the crack size. In detail, the internal pressure increasing factor *k* of the crack increases linearly as the crack size increases. The crack is in danger when the internal pressure increasing factor *k* of the crack is negative. That is, the pipeline crack will expand as the internal pressure increases.

## Figures and Tables

**Figure 1 sensors-20-00810-f001:**
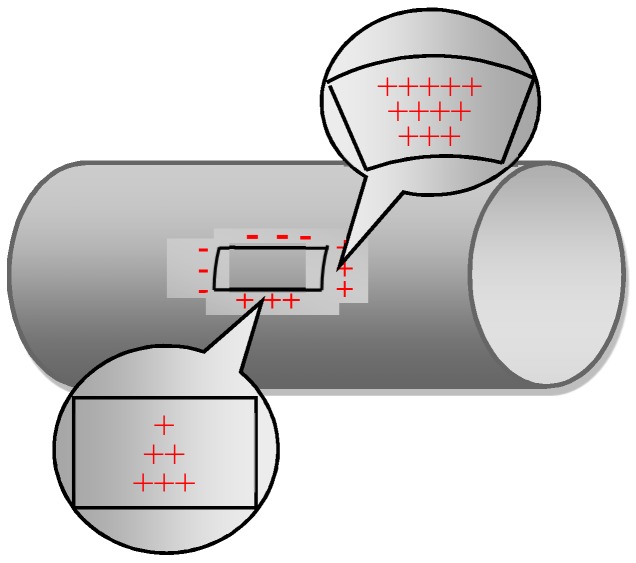
Schematic diagram of pipeline crack magnetic charge model.

**Figure 2 sensors-20-00810-f002:**
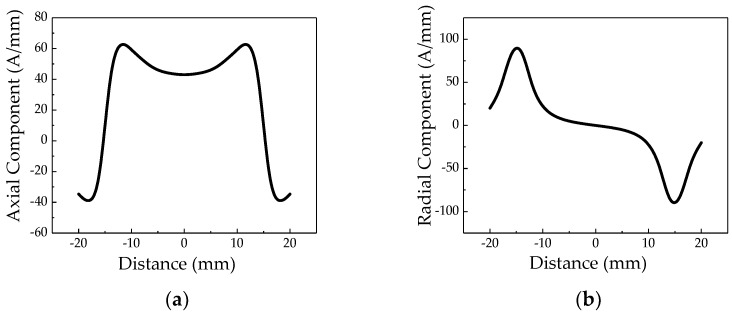
The characteristics of weak magnetic signal in model of non-uniform crack: (**a**) axial component and (**b**) radial component. The axial component of the weak magnetic at the crack has a maximum value near the crack tip, and a minimum value appears in the middle of the crack. The radial component of the weak magnetic at the crack has a peak and a valley at the crack tip.

**Figure 3 sensors-20-00810-f003:**
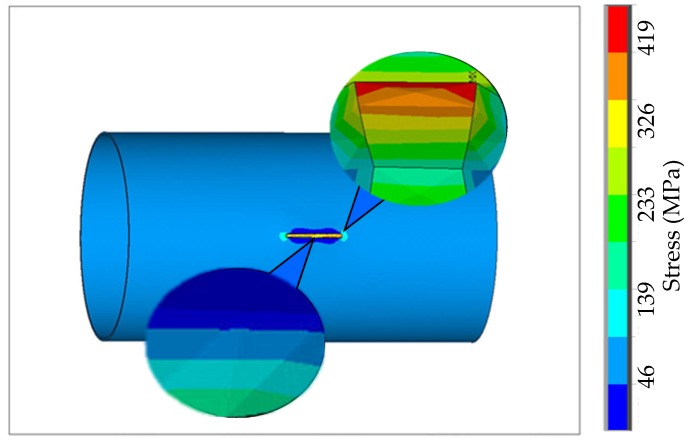
The stress diagram of pipeline crack.

**Figure 4 sensors-20-00810-f004:**
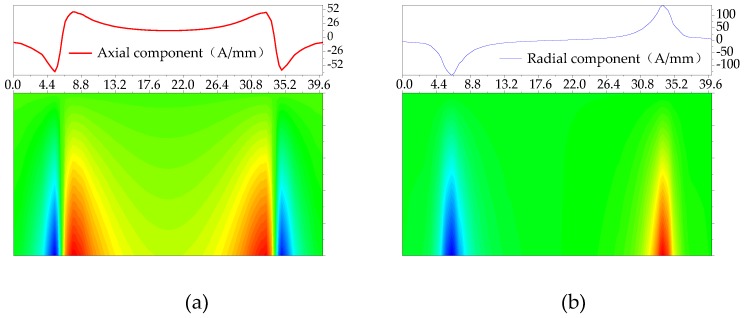
The characteristics of weak magnetic signals at the crack: (**a**) axial component and (**b**) radial component.

**Figure 5 sensors-20-00810-f005:**
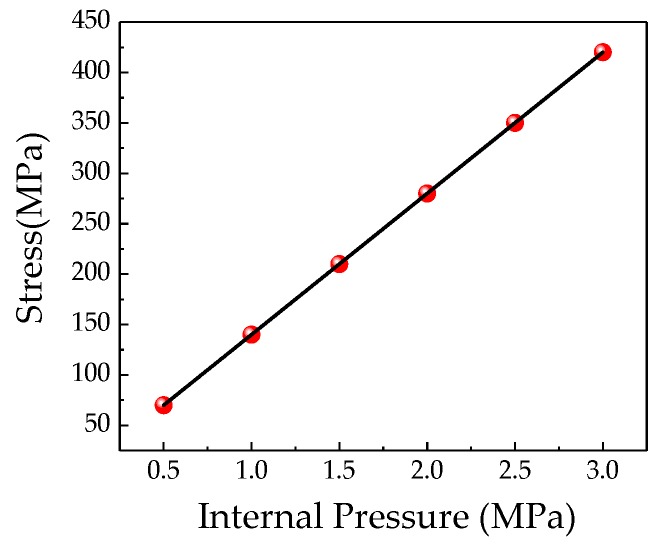
Relationship between the stress at crack tip and the internal pressure.

**Figure 6 sensors-20-00810-f006:**
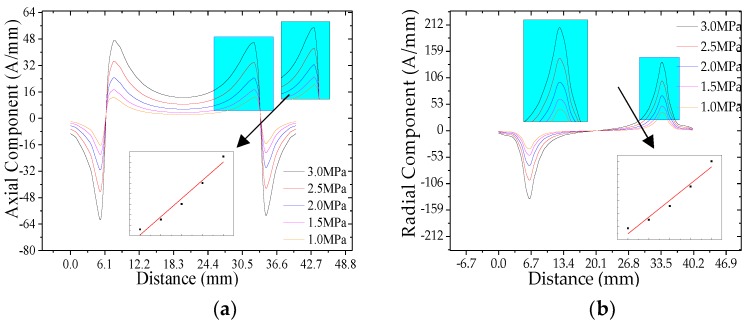
The weak magnetic signal of pipeline crack with different internal pressures: (**a**) axial component and (**b**) radial component. The axial maximum and the radial peak of the weak magnetic signal at the crack are extracted, and the relationship between the signal and the internal pressure is shown in the figure.

**Figure 7 sensors-20-00810-f007:**
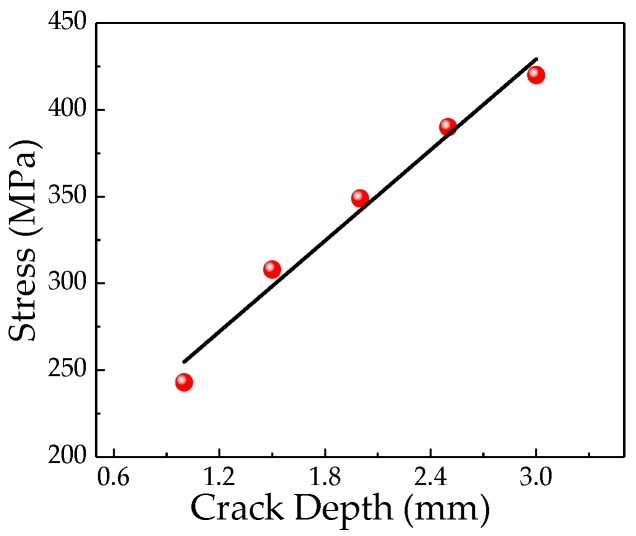
Relationship between the stress at crack tip and the crack depth.

**Figure 8 sensors-20-00810-f008:**
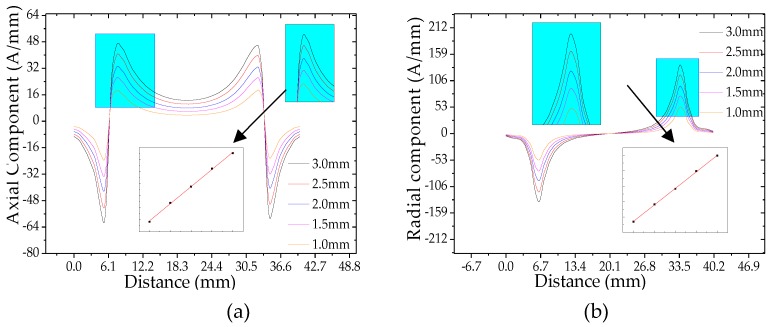
The weak magnetic signal of pipeline crack with different crack depths: (**a**) axial component and (**b**) radial component. And the axial maximum and the radial peak of the weak magnetic signal at the crack are extracted, and the relationship between the signal and the crack depth is shown in the figure.

**Figure 9 sensors-20-00810-f009:**
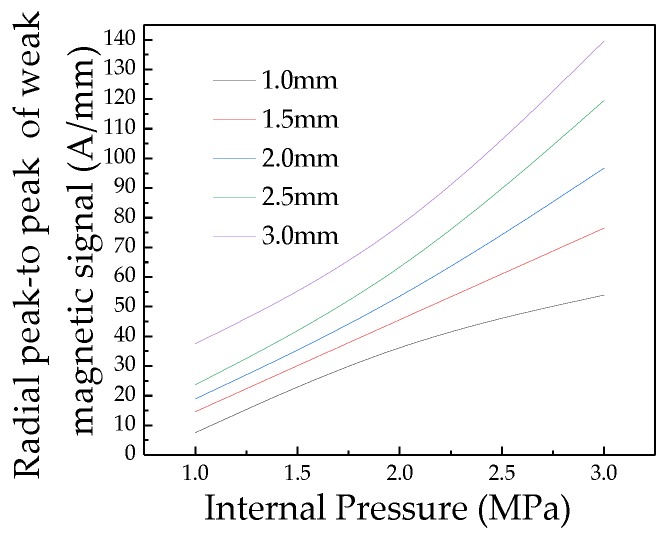
Radial peak-to-peak of weak magnetic signals of different crack depths with internal pressure.

**Figure 10 sensors-20-00810-f010:**
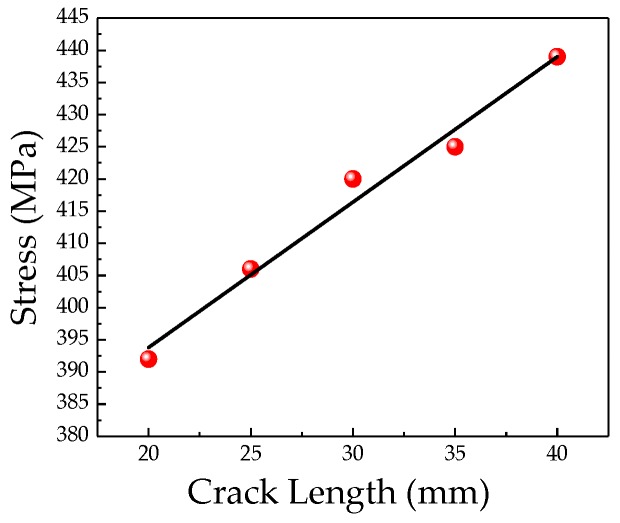
Relationship between the stress at crack tip and the crack length.

**Figure 11 sensors-20-00810-f011:**
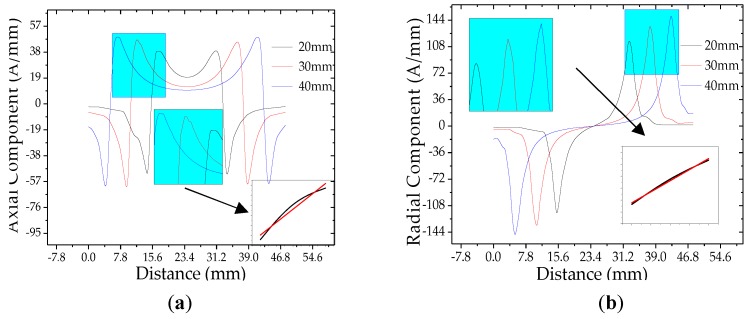
The weak magnetic signal of pipeline crack with different crack lengths: (**a**) axial component and (**b**) radial component. And the axial maximum and the radial peak of the weak magnetic signal at the crack are extracted, and the relationship between the signal and the crack length is shown in the figure.

**Figure 12 sensors-20-00810-f012:**
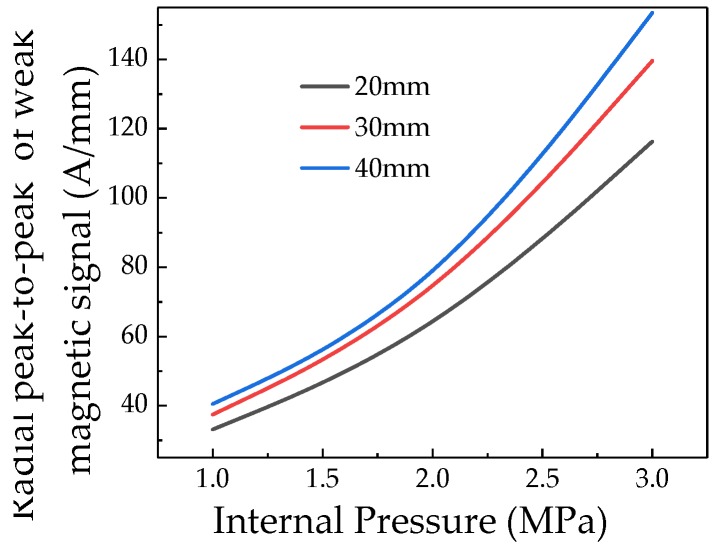
Radial peak-to-peak of weak magnetic signals of different crack lengths with internal pressure.

**Figure 13 sensors-20-00810-f013:**
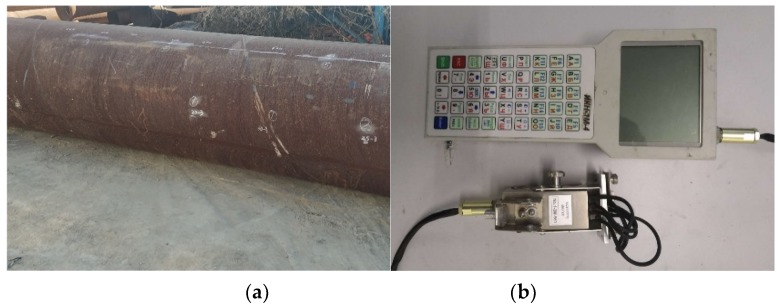
The experimental pipeline and weak magnetic detection device: (**a**) the experiment pipeline and (**b**) the weak magnetic detection device.

**Figure 14 sensors-20-00810-f014:**
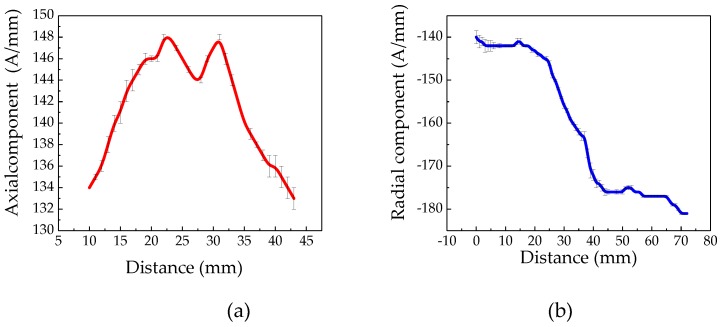
The characteristics of weak magnetic signal at the crack: (**a**) axial component and (**b**) radial component. An error bar illustrates standard deviation for the standard signal.

**Figure 15 sensors-20-00810-f015:**
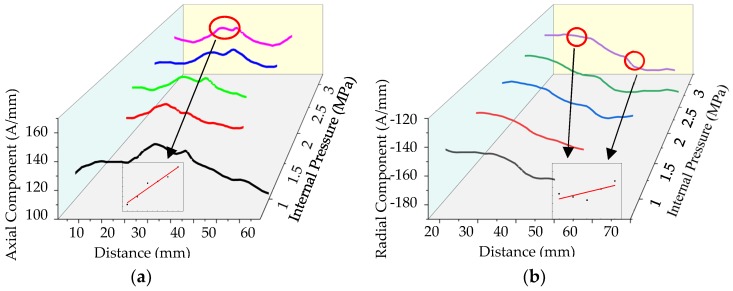
The weak magnetic signal of pipeline crack with different internal pressures: (**a**) axial component and (**b**) radial component. And the axial maximum and the radial peak of the weak magnetic signal at the crack are extracted, the relationship between the signal and the internal pressure is shown in the figure.

**Figure 16 sensors-20-00810-f016:**
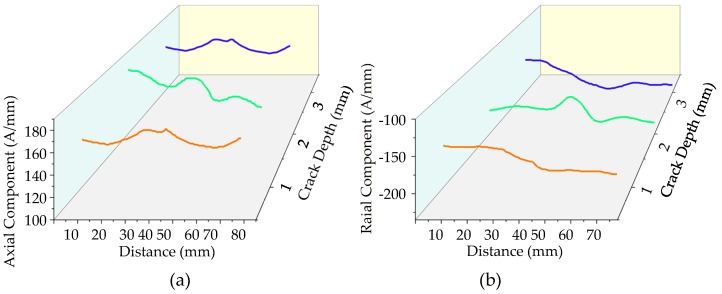
The weak magnetic signal of pipeline crack with different crack depths: (**a**) axial component and (**b**) radial component.

**Figure 17 sensors-20-00810-f017:**
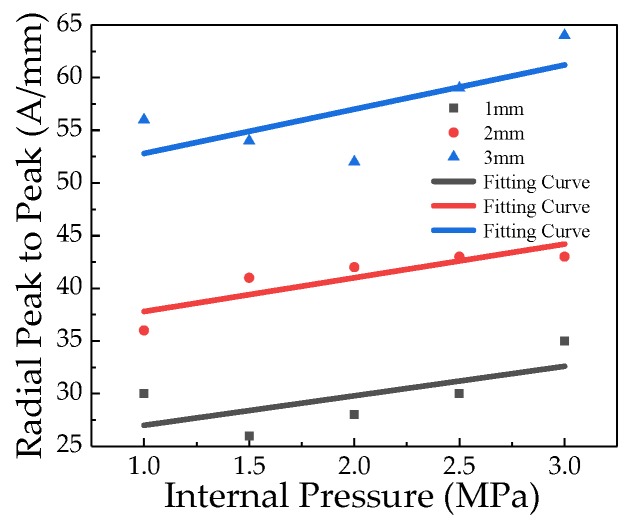
Radial peak-to-peak of weak magnetic signals of different crack depths with internal pressure. And the deeper the crack depth is, the faster radial peak-to-peak of magnetic weak signal changes with the internal pressure.

**Figure 18 sensors-20-00810-f018:**
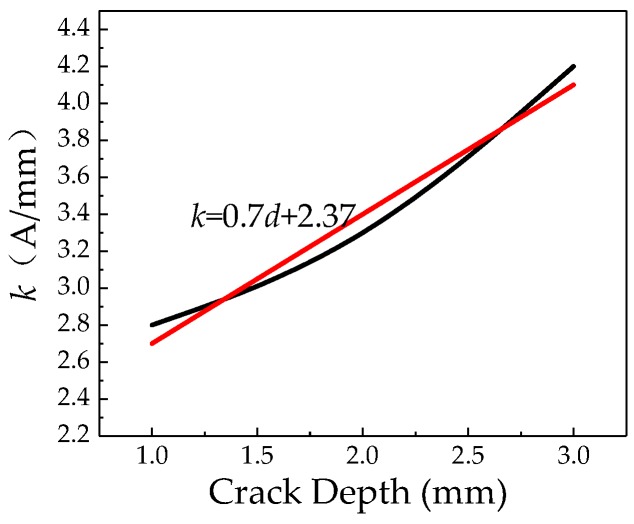
Relationship between the internal pressure increasing factor *k* and the crack depth.

**Figure 19 sensors-20-00810-f019:**
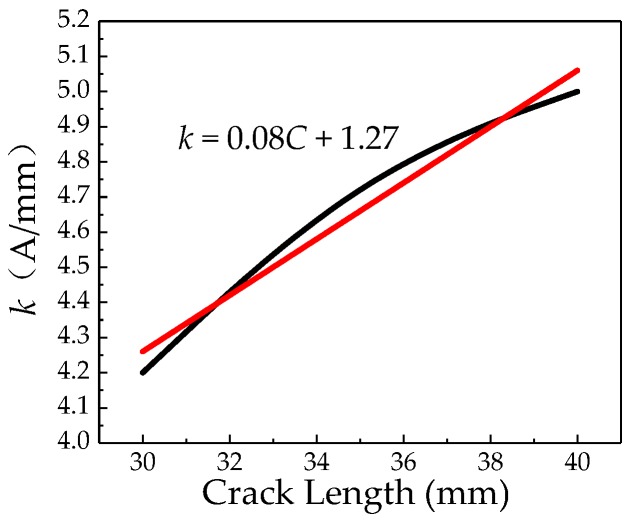
Relationship between the internal pressure increasing factor *k* and the crack length.

**Figure 20 sensors-20-00810-f020:**
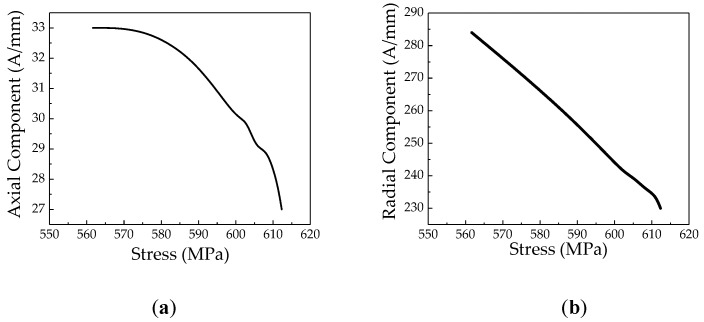
The weak magnetic signal change after the crack expands: (**a**) axial component and (**b**) radial component.

**Table 1 sensors-20-00810-t001:** Numbering table for cracks of different sizes.

Number	Length (mm)	Width (mm)	Depth (mm)
1	30	1	1
2	30	1	2
3	30	1	3
4	35	1	3
5	40	1	3

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
