# Peer review of "Research on Features of Pipeline Crack Signal Based on Weak Magnetic Method"

_sensors, 2020, doi:10.3390/s20030810_

Round 1

Reviewer 1 Report

This paper focus on quantitative online detection of microcracks in long-distance oil and gas pipelines by Weak Magnetic Method, which is interesting and important. There are some comments:

The authors need to check the grammar. For example, in line 136 “mpa” should be MPa. There is no unit in Figure 3 and Figure 4. Please explain the way to obtain the Figure 20 by experiment. How to decide the the crack begins to expand or not in the experiment. please discuss the difference between the simulation figure 4 and experiment figure 14.

Author Response

请参阅附件。

Reviewer 2 Report

The authors present an interesting study on features of pipeline cracks. The structures of the paper is clear, and the results are supported by the analysis. The figures are also very descriptive.

My major concern is related to the equations. This is not a technical issue, so I am not saying that there is a lack in the experiments. For instance:

(1) In equations (8) and (9), I would use the symbol of the square root rather than the power to 1/2. It is exactly the same thing, but with the square root symbol, it is easier to understand.

(2) I do not understand where the 1.16 in equation (9) comes from. Is this number the decimal approximation of a real number?

(3) Be careful with the round brackets in equation (3) because they should cover the whole expression.

(4) There is a general problem with the font size of the equations. A clear example is equations (12) and (13), where it can be clearly observed that rho_1 is greater than rho_2. A similar problem occurs in equations (14), (15) and so on, where the symbol of the integral is too small with respect to the integrand.

(5) In equation (20), the authors should not use a 'star' (*) as a symbol for the product. They should use the centered dot instead.

(6) The reference section should be improved. I do not understand the '[J]' after the name of some articles (references [2] and [12], for instance). Reference [24] has a family name in capital letters. In general, the references are not uniform.

Reviewer 3 Report

Please find my comments in the report attached.

Round 2

Reviewer 3 Report

The authors have addressed my concerns and replied satisfactorily to my comments and questions.

Author Response

Thanks for your approval, the manuscript has been modified in detail. And the errors in the article have been corrected.